# Iterating toward change: Improving student-centered teaching through the STEM faculty institute (STEMFI)

**Jeffrey Shipley[1], Rebecca L. Sansom[2], Haley Mickelsen[1], Jennifer B. Nielson[2], R. Steven Turley[3], Richard E. West[4], Geoffrey Wright[5], Bryn St. Clair[6], Jamie L. Jensen[1]***

**1** Department of Biology, Brigham Young University, Provo, UT, United States of America, **2** Department of Chemistry and Biochemistry, Brigham Young University, Provo, UT, United States of America, **3** Department of Physics and Astronomy, Brigham Young University, Provo, UT, United States of America, **4** Department of Instructional Psychology and Technology, Brigham Young University, Provo, UT, United States of America, **5** Department of Technology and Engineering Studies, Brigham Young University, Provo, UT, United States of America, **6** Department of Plant and Wildlife Science, Brigham Young University, Provo, UT, United States of America

* Jamie.Jensen@byu.edu

**Data Availability Statement:** Data are available from the Brigham Young University Institutional Scholars archive (via https://scholarsarchive.byu.edu/data/49).

## Abstract

One of the primary reasons why students leave STEM majors is due to the poor quality of instruction. Teaching practices can be improved through professional development programs; however, several barriers exist. Creating lasting change by overcoming these barriers is the primary objective of the STEM Faculty Institute (STEMFI). STEMFI was designed according to the framework established by Ajzen's Theory of Planned Behavior. To evaluate its effectiveness, the Classroom Observation Protocol for Undergraduate STEM (COPUS) tool was used before and after an intensive year-long faculty development program and analyzed using copusprofiles.org, a tool that classifies each COPUS report into one of three instructional styles: didactic, interactive lecture, and student-centered. We report the success of our program in changing faculty teaching behaviors and we categorize them into types of reformers. Then, thematically coded post-participation interviews give us clues into the characteristics of each type of reformer. Our results demonstrate that faculty can significantly improve the student-centeredness of their teaching practices in a relatively short time. We also discuss the implications of faculty attitudes for future professional development efforts.

## Introduction

Economic forecasts suggest that the demand for Science, Technology, Engineering, and Mathematics (STEM) majors is likely to increase by 5–20% justifying a need to increase retention of STEM majors [1]. Poor STEM teaching is a major contributing factor to attrition from STEM majors [2]. STEM classes are frequently taught didactically through lecture [3], which can cause students to disengage or struggle to learn while in class [4]. In contrast, we suggest students should be engaged in active and inquiry-based approaches, which include collaborative

**Funding:** "RLS, DMW, BES, REW, and JEJ were supported under grant DUE-1712056 from the US National Science Foundation (www.nsf.gov). Any opinions, findings, and conclusions or recommendations expressed in this material are those of the author(s) and do not necessarily reflect the views of the National Science Foundation. The funders had no role in study design, data collection and analysis, decision to publish, or preparation of the manuscript."

**Competing interests:** The authors have declared that no competing interests exist.

learning and student-centered teaching (SCT) strategies. Active learning may include dialoguing, group work, guided inquiry, or the use of personal response systems, among others (see Freeman et al. [5], for a meta-analysis). We specifically focused this faculty development on student-centered teaching strategies that we define as those that encourage students to be engaged in the learning process instead of sitting passively in class. The use of SCT strategies has resulted in several academic benefits including improved critical thinking skills, greater involvement of students in the learning process, and the personalization of large lectures [6]. Additionally, these strategies can improve student grades and achievement [5] and reduce the high attrition rates in STEM courses.

Faculty development workshops have emerged as a primary vehicle by which administrators and leaders in STEM education attempt to improve teaching [e.g., 7–9]. Such workshops often last for several hours a day over days, within and cross-discipline [10], and address a variety of topics including the importance of active learning to improve student understanding, engagement, and experience. Faculty participate in teaching workshops for a variety of reasons including discontent with teaching practices, student participation, or student experience [11].

Some faculty development programs have proven to be effective at enhancing faculty knowledge, professional competence, and student performance [4, 12]. However, such development programs frequently do not cause lasting changes to teaching strategies or student engagement. Several factors have been proposed as barriers to lasting change: (a) lack of awareness of the evidence that supports the use of SCT techniques [13], (b) reluctance by faculty to buy into the published literature since they frequently did not learn through SCT techniques themselves [13], and (c) inadequate follow-up after workshop participation to support implementation [14].

Past research has clarified some of the barriers to and drivers for instructional change [15, 16]. Baker et al. [7], suggest the need to align the framework of faculty support that includes institutional and department-level affairs, as well as individual instructor characteristics. A culturally appropriate context, particularly discipline-specific application on teaching scholarship is also desired [16] as is continuity in training within departmental cohorts [10]. Furthermore, empirical evidence beyond self-reported qualitative data will elucidate the impact of faculty development activities on student learning [17].

To address these barriers, the STEM Faculty Institute (STEMFI) program was created as a year-long faculty development program with a dual purpose to support lasting faculty change to SCT and to better understand what drives faculty to make that change.

## Theoretical rationale

We used the Theory of Planned Behavior as a theoretical framework to study the causal mechanisms involved in promoting lasting changes in STEM instruction. Originally proposed as a way to think about changes in public health behaviors [18], it addresses three different factors —attitude toward the behavior, subjective norms, and perceived behavioral control—that influence the intention of an individual to behave a certain way (see Fig 1). When an individual develops a favorable attitude toward a behavior, believes the behavior is expected by others and perceives the behavior as possible, the person's intention to perform the behavior increases [19]. We believe that these same principles apply to changes in teaching behaviors and thus we sought to design our faculty institute to address these factors.

The first factor, attitude toward the behavior, is an individual's overall perception of the behavior, often evaluated on the potential benefits or drawbacks to both the individual performing the behavior and others. In the context of STEMFI, our research sought to understand what attitudes faculty members had regarding the usefulness of SCT strategies and their

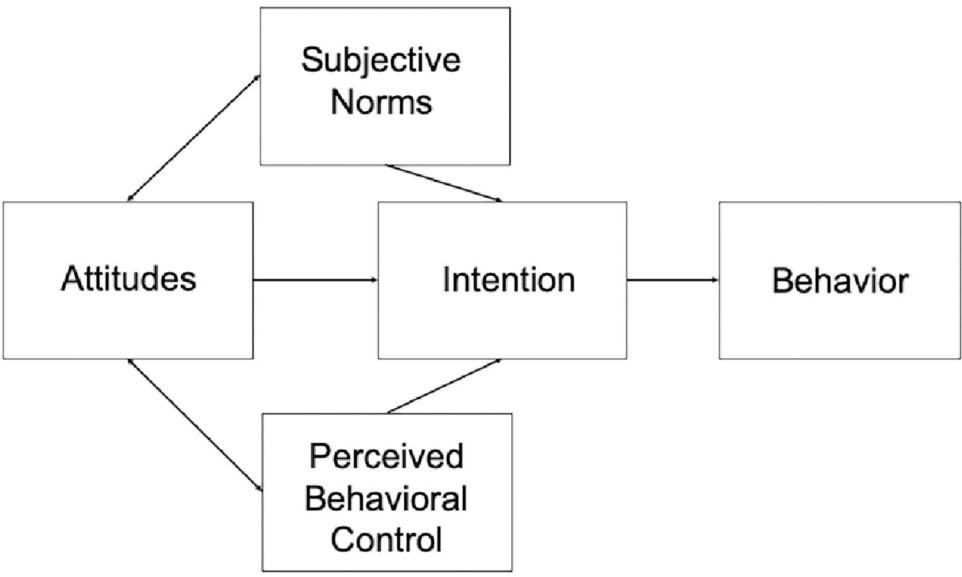

**Fig 1. The theory of planned behavior.**

effectiveness in helping students learn and stay engaged in STEM classes. Our program sought to support this attitudinal change by introducing faculty participants to the evidence supporting the use of SCT and structuring the workshop so that participants could experience the strategies for themselves and change their perceptions of the utility of SCT.

The second factor, subjective norms, "consist of a person's beliefs about whether significant others think he or she should engage in the behavior" and can hold sway over an individual's intention to carry out a behavior [19, p. 585]. We learned about participants' social experiences within their departments and colleges through pre-participation interviews. We structured the STEMFI program to promote positive subjective norms by providing opportunities to interact as a cohort and with a supportive mentor. At monthly cohort meetings, participants presented strategies they had employed and heard from others in the group about their chosen strategies. Together, they examined what worked and what did not work, and were encouraged in their quest to create a student-centered classroom—thus improving their subjective norms.

The final factor, perceived behavioral control, is a person's belief that they are capable of performing a behavior in their current situation; it is influenced by both their self-efficacy and external factors [19]. In our pre-interviews with faculty, we asked about the specific challenges they face or anticipate facing while implementing SCT. During the STEMFI workshop, participants received training on the use of SCT practices to support self-efficacy. We did not attempt to change external factors, like classroom setup, but we did try to help participants see how those challenges could be overcome.

To incorporate each of these factors, the STEMFI program was designed as a year-long program that began with a week-long summer workshop where participants experienced SCT strategies, learned about the evidence from discipline-based education research [20] to support the use of those strategies, and built a social network with colleagues. We aimed to answer the question, can we promote reform of teaching to more SCT through the lens of the Theory of Planned Behavior? By comparing teaching observations from the two semesters following the week-long summer workshop to the pre-workshop data and interviewing faculty at the end of the program, we were able to directly measure changes in faculty teaching and attitudes and therefore understand the effectiveness of the STEMFI program in facilitating those changes.

## Materials and methods

### Ethics statement

All participants provided consent to participate in the research study. Permission was obtained from the primary authors Institutional Review Board, approval number X17244.

### Participants

The STEMFI program was established with a National Science Foundation grant at a large, private doctoral-granting institution in the western United States. Approximately 35,000 students attend the institution, and 12,000 are enrolled in a STEM degree program. Approximately 51% of the student body is female, 77% single, 81% Caucasian, 9% Hispanic or Latino, and 1% black students.

The STEMFI program was run in three year-long cohorts, consisting of 15 faculty each, over the course of four years (a gap year occurred due to COVID). Faculty came from the three STEM colleges on campus for a total of 45 faculty: 19 from Life Sciences, 13 from Physical and Mathematical Sciences, and 13 from Engineering and Technology. The Colleges were nearly evenly represented in each cohort. Eighteen participants were Assistant Professors (pre-tenure), 21 were Associate Professors (tenured), and six were Full Professors (the highest rank obtained post-tenure). There were 35 males and 10 females that participated, which, at this institution is an overrepresentation of females when compared to the faculty at large. All faculty participants volunteered to participate and were compensated with a small stipend ($600) to their research account. Each received approval from their respective department chair and dean.

### STEMFI program

The STEMFI program lasted three to four semesters (over two years) and consisted of three phases: pre-, during, and post-workshop. Pre-workshop observations using the Classroom Observation Protocol for Undergraduate STEM (COPUS; Smith et al. [21]) were performed for all participants during the first year. A goal of four class-length observations were made for each faculty participant (although some received only three observations). The observations were performed on random days, and as often as possible, were observations of the same course that they planned to reform during STEMFI. We also conducted pre-workshop interviews in order to better tailor the workshop experience. Participants did not reform during the first year.

Phase 2 consisted of faculty participation in a one-time summer workshop lasting one full week (9am - 5pm), where we worked to improve their attitudes, subjective norms, and perceived behavioral control specifically for SCT through an active learning experience, collaboration with colleagues, and focused instruction on strategies. Several student-centered strategies were introduced to encourage instructors to facilitate a more active classroom. While we recognize that active learning is not always student-centered, and our COPUS instrument focused on active learning, the workshop specifically focused on student-centered strategies. The workshop is described in detail in West et al. [22]. They were required to complete one fully reformed lesson plan that specifically included SCT and encouraged to tackle a second during the week. Participants were assigned a peer mentor. In the first cohort, the workshop facilitators, along with a few other faculty who were chosen for their excellent teaching record, served as mentors. In the second and third cohorts, we chose participants from the previous cohort who had demonstrated significant reform. We tried to pair mentors and participants who were in the same discipline (so that they understood the disciplinary nuances) but outside of

the department (so that they had no influence on rank or status decisions). During the workshop, participants also met with their peer mentor and made plans for implementation. The participant experience and workshop design are described in more detail in a recent publication [22].

In phase three, we followed faculty participants for two semesters after the summer workshop (one full academic year). During the first semester they taught following the summer workshop, participants were encouraged to add at least three new SCT techniques to their teaching. Some faculty chose simpler strategies such as student response systems (clickers) or think-pair-share [e.g., 23], while others chose more involved strategies like the 5E learning cycle [24], Process-oriented guided inquiry learning (POGIL) [25], or Decision-based Learning [26]. Participants also met regularly with their mentors to practice new strategies, discuss previous efforts, and set goals—actions that can be helpful in supporting lasting change [22]. At least three (with a goal of four) of these classes were also observed using COPUS to measure their post-teaching behaviors. In addition to the one-on-one guidance from an individual mentor, participants also received social support from colleagues in the cohort at monthly cohort meetings for a full academic year where they shared what they had done and brainstormed ways to improve or apply the strategy to a different course [22].

## Observation instrument for quantitative analysis

The COPUS tool (Smith et al. [21]) was used to gather quantitative data. All data is available at https://scholarsarchive.byu.edu/data/49. COPUS is a data collection tool that records student and instructor behaviors every two minutes during a given class to assess active interaction. Codes include more teacher-centered approaches, such as lecturing by the instructor and listening by the students, more interactive strategies like student questions and instructor answers, and more fully student-centered strategies like group work, clicker questions, and the instructor moving and guiding around the room. It does not, however, assess specific student-centered techniques. For more description of the instrument and codes, see Smith et al. [21]. By documenting the activity of the students and teacher with a variety of codes, the observer can measure the level of student engagement and infer the degree to which the classroom is active. Undergraduate and graduate students were trained to use the COPUS using the training protocol established in Smith et al. [21]. Participants were observed in person three to four times prior and four times after (the switch from 3 to 4 occurred between cohort 2 and 3, being informed by the intervening publication of Stains et al. [3]). Pre-observations were taken at random to try to capture typical class periods. Participants were made aware of the observation, but the observer was usually a student who blended in with the class. Post-observations were made in the first semester they taught following their participation and were selected by the participant in order to showcase the new techniques they were planning to use. Thus, the post-observations represent what the participants felt were most representative of what they had learned and chosen to implement based on their STEMFI experience, and were not random.

Of our 45 participants, four were unable to complete the program, one due to COVID class cancelation, two due to unexpected leaves, and one due to his course being entirely online and inaccessible. An additional three participants had significant shifts in their course structures, due to COVID, such that post-data was collected on hybrid or online courses, but their data was still obtainable and included in analysis. These instructors provided recordings of their classes that they conducted in a hybrid or online fashion and we used the COPUS to analyze behaviors. Certainly, the online conditions hampered some active learning strategies causing measure of reform to likely be lower than they would have been in person.

## Classification of participant classroom behavior

After a nationwide study and latent class analysis on more than 2,500 classroom observations, Stains et al. [3] created an online tool to classify instructor practice as didactic, interactive lecture, or student-centered at copusprofiles.org. As Stains et al. [3] describe, didactic "depicts classrooms in which 80% or more of class time consists of lecturing"; an interactive classroom "represents instructors who supplement lecture with more student-centered strategies such as 'Other group activities' and 'Clicker questions with group work'; and student-centered instructors "incorporate student-centered strategies into large portions of their classes" (p.1469). We used this tool to classify each COPUS observation for each participant. Participants were classified based on the majority (two or more) of their observations. For example, if a participant had four pre-observations labeled as didactic, didactic, didactic, and interactive, they were labeled as "didactic".

Participants who moved from didactic to interactive were labeled as "Beginning Reformers"; those who moved from didactic all the way to student-centered were labeled as "Dramatic Reformers"; those who were already interactive and moved to student-centered were labeled as "Advanced Reformers"; those who were already using interactive strategies and remained interactive (although with broadened strategies) were labeled as "Interactive Reformers"; likewise those who were already using student-centered strategies and continued being student-centered (with broadened strategy use by trying new strategies that they learned in the workshop) were labeled "Student-Centered Reformers"; and lastly, those who started with and chose to continue with only didactic strategies, even after the intervention, were labeled as "Didactic Non-Reformers" (see Fig 2).

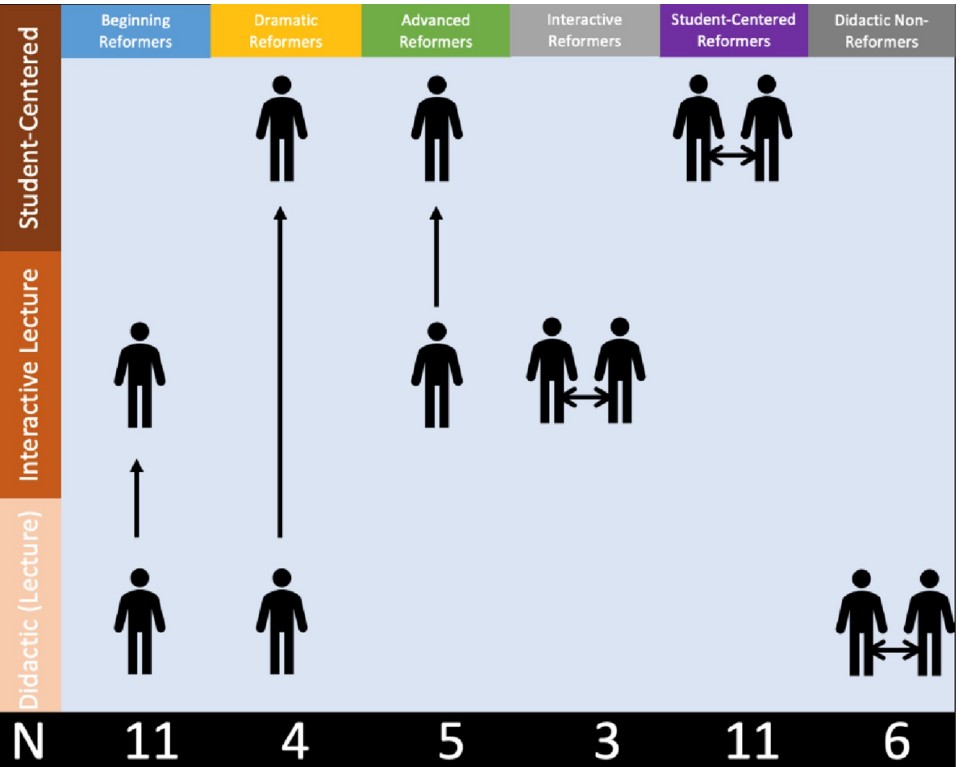

**Fig 2. Types of reformers based on movement between groups.** The "N" indicates the number of participants in each category.

**Interview protocol for qualitative analysis.** At the end of their STEMFI experience, we interviewed participants. The interview protocol, included in the supplementary information, addressed some programmatic evaluation pieces (e.g., Which STEMFI activities were most/least helpful?) and the three factors of the Theory of Planned Behavior (e.g., for Attitudes, we asked questions like, Have your attitudes about SCT changed over the course of your participation? For Subjective Norms, we asked questions like, How have your students responded to the changes you've made? For Perceived Behavioral Control, we asked questions like, Has your confidence to use SCT changed?). The full interview protocol is included in Supplementary Materials. Full transcripts are available at https://scholarsarchive.byu.edu/data/49.

Interview transcripts were read and thematically coded by JS, HM, and JLJ following the protocol outlined by Strauss and Corbin [27]. After the first reading of the interviews, readers compiled lists of themes, backed with quotations, that emerged from the interviews. Such themes were discussed and combined into four main themes that described issues relevant to participants' decisions to reform their teaching. Within these themes, we made binary categories into which we put each participant. Each interview was then recorded into these emergent theme categories.

In cases where the two independent researchers did not agree on the appropriate categorization for a participant's interview, a third researcher also read and interpreted the transcript. The three researchers then met to discuss the textual evidence that supported their ratings and continued the discussion until consensus was reached.

We then used an explanatory mixed methods design [28] to merge the findings from the quantitative and qualitative data. In the process of merging the quantitative and qualitative data, we used the Theory of Planned Behavior to organize and provide context for our findings. However, we found that the themes were not robustly tied to specific reformer types in an exclusive way, so they can only hint at potential differences between reformers.

## Results

### Quantitative results

Of the 41 participants with complete data, we classified 35 as "reformers" because they were able to "reform" their teaching in some significant way by successfully implementing more student-centered practices. These changes revealed that 85% of our participants were able to improve their teaching, suggesting that the STEMFI program was effective in changing faculty behavior.

Of these 35 reformers, 11 were classified as beginning reformers who moved from didactic instruction to interactive lecture (two Assistant and nine Associate professors), four as dramatic reformers who moved from didactic instruction to student-centered strategies (three Associate and one Full professor), five as advanced reformers who moved from interactive lectures to student-centered strategies (one Assistant, three Associate, and one Full professors), three as interactive reformers who already used interactive lecture techniques and simply incorporated more or different interactive lecture techniques (one Assistant, one Associate, and one Full professors), and 11 as student-centered reformers who were already using student-centered strategies and simply added new and different strategies to their repertoire (nine Assistant and two Associate professors). Only six faculty participants failed to move beyond didactic strategies (four Assistant and two Full professors) (see Fig 2).

### Qualitative results

Through inductive thematic analysis of post-experience interviews, four themes emerged that seemed to be influential in participants' decisions to make changes to their teaching practices: (1) attitude toward SCT, (2) student responses to SCT, (3) participant motivation, and (4) challenges. We then created dichotomous categories within each theme. Through the lens of the

Theory of Planned Behavior, we triangulated these categories with COPUS data to make loose hypotheses about the motivations of each reformer. However, we found that the themes were not robustly tied to specific reformer types in an exclusive way, so they can only hint at potential differences between reformers.

## Attitudes toward student-centered teaching

Attitudes toward SCT were categorized as either fully reformed or in transition. Fully reformed individuals displayed attitudes that indicated buy-in to the idea of SCT being beneficial and more effective than the traditional lecture-style approach. For example, one participant commented,

> I have a lot more confidence in knowing that this is a good way to use class time and. . .seeing them all working, trying to figure out what the answer is. . . it gives me a lot of confidence that [SCT] really is a worthwhile thing.

Most participants in the program displayed fully reformed attitudes, especially among those who were primarily didactic in their approach to teaching (i.e., beginning and dramatic reformers). We also see this attitude among those who were already well-versed in student-centered strategies (i.e., student-centered reformers). In the framework of the Theory of Planned Behavior, these findings are associated with favorable attitudes toward SCT consistent with a self-directed choice to participate in the workshop.

In contrast, some participants, classified as in transition, made comments that seem to indicate they had some reservations about SCT, while maintaining an overall positive attitude. For example, one participant commented that SCT

> was really interesting and the students were very involved, but I always think about, well, what do you do on the days where it's not as interesting? 'Cause, there's some hard days when the topic's just not going to entice the students quite as much and so. . . I don't know,

indicating that he was not sure that SCT would fit for all content. Another participant, while talking about creating a SCT activity for a particular lesson, stated,

> A whole day is that one [SCT activity], and I'm not prepared to say that we can afford to do that, or if it would be better but we won't have time. That's something that I don't know the best answer for,

indicating that she appeared to be unsure whether the time spent doing SCT was worth the benefits. We saw these attitudes mainly among those who implemented some reformed strategies prior to the workshop, but who were not fully engaged in SCT (i.e., advanced reformers), and among those who chose to make no changes to their teaching (i.e., non-reformers).

## Student responses to student-centered teaching

Student response to the implementation of SCT techniques in the classroom was categorized as positive or negative. Positive feedback would indicate that the students enjoyed the new teaching style, or saw the benefit in it. For example, one participant commented,

> There was a positive impact from those newly implemented activities in my classes. Student ratings went up, and so I had the highest rating I ever had in that class. I have been teaching that class for five years now. . .and it was the highest rating I ever got.

Another participant noticed a positive change in student behaviors, "For the next three or four class periods, [the students] were more willing to ask questions [and be] more openly engaged." In the framework of the Theory of Planned Behavior, positive feedback from students contributed to favorable subjective norms, where student evaluations of teaching and student feedback are viewed as extremely important indicators of the social context that faculty members experience. Those who began as didactice lecturers seemed to experience only positive feedback as they took their first steps into SCT (i.e., beginning and dramatic reformers). Others had mixed feedback.

Negative feedback would indicate that students were resistant to or expressed their dissatisfaction with the activities and could be seen as a significant challenge to participants. For example, one participant noted that "after four days of doing it, I asked them, 'Did you like this lesson model?' No. [They] did not like it." Another participant noted a lack of student buy-in to the activities saying,

> I felt like 'Oh, I'm coming in with these ideas and I have more student engagement things than I've had before, and this should be really cool,' and they just didn't seem to buy it or buy into it, it just. . . I don't know.

Most of the participants who were attempting to implement more advanced SCT strategies experienced at least some negative feedback (i.e., advanced and student-centered reformers), however, they demonstrated overwhelming positivity and a desire to continue using reformed strategies. This was especially true among student-centered reformers who used negative feedback as a motivator to be even more engaging, more open, more welcoming, as is seen in this comment:

> I think one of the biggest challenges was the student engagement, or lack thereof, and I have not figured out how to overcome that. . .. Last semester was just really rough. . . there were lots of times where I was like, "Ugh, I have to go to this class". . . I don't know how you make it. . . more engaging, make it more open, make it more welcoming, I'm not really sure."

This participant's commitment to remain student-centered despite the struggles with negative student reactions stands out as a characteristic of a student-centered reformer. In contrast, we also see negative student feedback in non-reformers experienced, which appeared to stifle their desire to change.

## Participant motivatio

Participant motivation was divided into intrinsic and extrinsic. All of our participants had an intrinsic drive to participate in STEMFI. Intrinsic motivation was characterized by faculty members who had a genuine desire to become a better teacher. When asked why they signed up for STEMFI, one participant commented,

> I always want to improve my classes, and I want to become a better teacher, and so by signing up for [STEMFI], I can go from just having that as an ideal to actually trying to put a plan into action.

Another said, "I'm always trying to come up with new. . . active learning experiences, activities to do with the students, and so I thought [STEMFI] would be really cool." In the

framework of the Theory of Planned Behavior, this motivation significantly contributed to their overall attitudes toward teaching reform.

Occasionally, a faculty participant would express some extrinsic motivation characterized by participation in the program for reasons external to an innate desire to be a better teacher. For example, one faculty member commented that she signed up because

> you [meaning the STEMFI team] invited me of course. [A colleague] told me I should do it, and I was free that spring. . . plus the NSF has the broader impacts part in our grants, so I was interested to see if I could tie something into that.

Another faculty member joined after being prompted by a Department Chair in preparation for a rank and status decision. This extrinsic motivation was particularly salient among non-reformers.

## Challenges

**Beyond the challenge of negative student feedback,** additional challenges faced by participants were categorized as logistical or philosophical. Logistical challenges were challenges that related more to time constraints, classroom architecture, the desire to cover course content. For example, one participant commented that "time management's the hardest thing, for sure." Another faculty member said,

> The timing is really hard because we have such a variety of students in the classroom. . . the people who don't have a physics background take a really long time to do stuff, and the people who do have a physics background take a very short time to do things and then you kind of have this. . . challenging situation.

Another commented on the classroom structure saying, "every desk is full, and they have all their stuff on the floor and I. . . have to walk across the front, but there's no way for me to get to the sides." In the framework of the Theory of Planned Behavior, these challenges were representative of their perceived behavioral control. All reformers expressed logistical challenges. Not surprisingly, the amount of logistical challenges mentioned by participants was somewhat related to the level of change they made. For example, beginning reformers, who only made small challenges, reported very few challenges. Dramatic reformers, on the other hand, reported significantly more challenges to their perceived behavioral control, particularly with regard to time in class and not knowing how to make SCT part of a fast-moving course.

Philosophical challenges, on the other hand, relate more to issues with SCT approaches, administrative pressure, department climate, or a lack of confidence. One faculty participant expressed a conflict between their own beliefs and SCT ideas saying,

> I'm kind of cautious in trying to introduce those things because I really do feel like as soon as you do those activities, you're giving up a significant amount of control over the time of the class—significant amount of control over what the students are "supposed to get" based on what I lecture. When I lecture you hear every word, and it goes into your brain and then it's there. That's not true, but that's the intention when I lecture. That's what I intend to have happen and then what I expect the students to have and I can justify moving onto the next topic because we already did the old thing, and that's false, but as soon as I do the opposite, which is, "We're going to let everyone float a little bit," then I lose that semblance of control and I feel. . . less at ease about doing that."

Another faculty participant expressed concern over not getting good student ratings for his/her rank and status portfolio:

> I'm like, "If I make these changes to the way I do exams and students hate it, then they're going to ding me on it and is that going to affect my ability to be promoted here?" And that's kind of not the way you should be thinking. You shouldn't be worrying about some other external pressure, right? So, that's what I mean when I think there's those kinds of external conflicts that are imposed that. . . maybe aren't ideal.

In response to department climate, one participant said, "I do not believe that the department encourages experimentation, exploration, and engagement on these type[s] of things [meaning SCT]." Several expressed a lack of confidence in using SCT techniques saying, "I value the time in class, I think it's sort of precious, and I get really nervous about trying to do new things because I don't want to fall flat, and I've seen a lot of lectures where you try to do fancy stuff and it's like you tried hard and it didn't work." Or, "I'm not good at it yet. So, it'll be just practicing and refining the technique of making. . .them discussing more than just me." These philosophical challenges were primarily seen among non-reformers, and surprisingly by those who were well-established as student-centered practitioners (i.e., student-centered reformers). However, student-centered reformers are set apart by how they perceived that challenge. Many participants in other profile groups indicated that using SCT meant they would not have time to cover all the material in their class, and viewed that as a flaw of SCT. In contrast, student-centered reformers were more likely to acknowledge their lack of confidence or skill at facilitating SCT, and to look for creative solutions to the problem rather than rejecting SCT outright. This philosophical challenge was caused by introspection, recognizing ways in which they could potentially improve as an instructor, and indicated their tendency to be more concerned with how well students were learning. Ultimately, student-centered reformers have such strong positive attitudes about SCT that they were able to overcome significant challenges related both to subjective norms and perceived behavioral control.

## Discussion

The STEM Faculty Institute was successful at creating change in at least some of the instructional behaviors of STEM faculty. By analyzing the post-interviews of these participants, we were able to characterize some of the attitudes of and challenges faced by participants and loosely relate them to reforming attitudes. While COPUS data could clearly differentiate levels of reform, the attitudes discovered in the interviews was not able to clearly differentiate between reformer types. However, the identified themes can offer insight into how attitudes, subjective norms, and perceived behavioral controls can influence desires to reform.

From beginning reformers, we learn several lessons. Beginning reformers chose to implement strategies that were simpler and less time-consuming, such as using clicker questions or think-pair-share activities, and perhaps that choice explains why they did not encounter significant challenges or barriers to the adoption of those strategies. This result may provide insight for future faculty development programs in that encouraging smaller changes to teaching behaviors was less likely to cause participants to encounter difficult challenges or barriers to implementation. Although it did not result in dramatic changes to teaching practice, the changes were measurable and received favorably by students. Additionally, beginning (and dramatic) reformers–those who came into the workshop with a didactic teaching style–showed fully reformed attitudes suggesting that the workshop was inspiring and motivating to them, significantly influencing their attitudes, which likely motivated their willingness to try new

strategies. Having a sufficiently interactive and enthusiastic workshop style may contribute to success.

Dramatic reformers, those who started as traditionally didactic instructors and implemented fully student-centered strategies, also provide valuable insight about effective professional development. One of the main themes in this group is that they experienced significant challenges. Because they were implementing dramatic changes to their curriculum, it is not surprising that their challenges were substantial. This can inform future faculty development programs by reminding us that those making dramatic changes are likely to require more scaffolding and support.

Advanced reformers are an interesting group. These are participants who were already using some reformed strategies, such as interactive lectures, who attempted to implement additional SCT strategies into their repertoire. One thing to note is that they expressed mixed attitudes toward SCT. It is possible that negative student feedback and their lack of expertise in SCT led them to question the true effectiveness of their reforming efforts. With more practice, it may be possible to shore up their attitudes about the importance and effectiveness of SCT. This can inform future faculty development efforts in reminding us that continued scaffolding and encouragement are likely important features of a successful reforming experience. Although advanced reformers report experiencing many logistical challenges, they are not deterred from implementing SCT. In fact, it may be due to their increased effort to implement SCT that such logistical challenges became apparent. Thus, when they encountered such challenges, they were motivated to overcome them. That dramatic and advanced reformers experienced similar logistical challenges may inform future faculty development programs. If a shift to SCT is intended to give students more control over their learning experience, departments may need to reevaluate their expectations for content coverage in favor of deeper learning of fewer topics.

Student-centered reformers were those who already had demonstrable experience implementing SCT strategies coming into this experience. We found that this group had fully reformed attitudes. Having extensive experience with these SCT strategies seems to have solidified their attitudes about the effectiveness of student-centered teaching despite any negative student feedback or significant challenges. Thus, in this group, focusing on convincing them that SCT is better than didactic strategies is likely not an effective use of time. Rather, further encouragement of their efforts is warranted. Student-centered reformers persevere in the face of negative student feedback, using it as a motivator for increased change, rather than as a stumbling block. They do not typically resort to lecturing even when SCT is challenging. Future professional development programs should understand that in the case of both advanced and student-centered reformers, faculty had to deal with mixed feedback from students. Their chosen degree of student-centeredness may depend on how committed they are to the philosophy of SCT.

Non-reformers displayed many of the same attitudes as those who chose to make measurable changes. First, non-reforms expressed a transitional view of SCT. In ways we do not yet fully understand, we were unable to significantly shift their attitudes and therefore their intentions to change their behaviors. This may have to do with their expressions of extrinsic motivation for participation (e.g., their Department Chair encouraged their participation, or they were participating in order to bolster their Broader Impacts section of NSF grants). Although professional development designers have little control over incoming attitudes, this stresses the importance of instilling positive attitudes about the benefits of SCT during the workshop. Additionally, while non-reformers did experience some positive feedback from students, they also experienced negative feedback. Because we did not quantify the amount or severity of student feedback, it is impossible to predict the exact effect of negative feedback on perceived

behavioral control. However, we can hypothesize that perhaps the negative feedback among non-reformers was more significant and impactful than among other reformer groups. We stress the importance of continued follow-up and effective mentoring to avoid these negative impacts.

It is worth considering certain limitations to our data. A small sample size (n = 41), due to limited resources, provides us with only preliminary conclusions. However, in the future, with the aid of additional resources, we hope to have larger cohorts of STEM faculty to better understand the effectiveness of the STEMFI program. Additionally, it is important to acknowledge that our study was conducted at a large, private university in the western US, which may limit us in applying our conclusions about overcoming barriers to faculty change to other learning environments. Another potential limitation is that our sample consisted mostly of volunteers who may have already had an increased interest in reforming their instruction, although a few faculty participants were strongly encouraged by the administration to participate because of low teaching evaluations. Certainly, a study where faculty were all compelled to participate could give more unbiased data about the program's effectiveness. Lastly, because our post observations were not chosen at random, but were chosen by participants to showcase the strategies they had learned and chosen to implement, the measure of change may be an overestimate of fully reformed teaching. In other words, the features observed after reform may not have been representative of the entire course or of lasting change. However, our goal was to motivate participants to use any SCT strategies, so we felt it was still representative of their use of strategies.

Based on the improvements in the teaching of the majority of our participants, we assert that the Theory of Planned Behavior was an effective framework for producing change in faculty behavior. Part of the STEMFI workshop directly addressed the effectiveness of SCT; we believe this played a role in shifting attitudes of our participants towards SCT and enabled them to internalize the belief that SCT was beneficial to student learning and development. Through monthly cohort meetings and the week-long summer workshop, we enabled faculty in cultivating positive subjective norms, meaning that they had regular interaction with other faculty that were trying to make the same difficult transition to more SCT. Regular meetings between mentors and participants were essential to effecting lasting teacher change. Faculty who receive regular mentoring report noticeable benefit to their teaching [29]. We recommend that additional studies employ a mentoring program to enable participants to implement the SCT strategies they have acquired.

Other studies on faculty development programs cite numerous barriers that impede faculty development. Satisfaction with current methods of instruction, such as traditional lecture without student involvement, is one such barrier [30]. Another faculty development program, called the Summer Teaching Institute, found that one and two years after the program, 98% of alumni said they were "still experimenting to improve their teaching"; however, a "lack of respect of colleagues in the department" was a major barrier to additional success in implementing SCT [9]. STEMFI sought to address this concern by emphasizing the mentor-participant meetings and monthly cohort meetings. Our results indicate that regular meetings with experienced mentors and other faculty engaged in implementing SCT were instrumental in aiding the majority of our STEMFI participants in their reform. However, we have not collected longitudinal data to test whether the change is lasting. Future studies of this methodology are needed to assess the perseverance of such change. Our preliminary study suggests that by providing instruction aimed at changing the participants' attitudes toward SCT in the summer workshop, improving their subjective norms through mentoring and regular cohort meetings, and helping faculty develop a positive view of their perceived behavioral control, we have built upon the efforts of previous faculty development programs to create sustainable and lasting change.

## Conclusions

Based on our data, we assert that teaching practices are malleable. As a professor develops a more positive attitude towards SCT, interacts with other faculty striving to do the same, and develops the intrinsic belief that he or she has the ability to implement such changes in the classroom, intention is refined, and behaviors are changed. As more faculty continue to develop an understanding of SCT and its benefits to students, we anticipate faculty overcoming barriers and implementing them in the classroom.

## Supporting information

**S1 File.**
(DOCX)

## Acknowledgments

We thank the undergraduate researchers who performed COPUS evaluations. We are grateful for those who helped organize and execute the summer workshop and the STEMFI participants for allowing us to observe their teaching.

## Author Contributions

**Conceptualization:** Rebecca L. Sansom, Jennifer B. Nielson, R. Steven Turley, Richard E. West, Geoffrey Wright, Jamie L. Jensen.

**Data curation:** Rebecca L. Sansom, Jennifer B. Nielson, R. Steven Turley, Richard E. West, Bryn St. Clair, Jamie L. Jensen.

**Formal analysis:** Jeffrey Shipley, Rebecca L. Sansom, Haley Mickelsen, Jennifer B. Nielson, R. Steven Turley, Richard E. West, Bryn St. Clair, Jamie L. Jensen.

**Funding acquisition:** Rebecca L. Sansom, Jennifer B. Nielson, R. Steven Turley, Richard E. West, Geoffrey Wright, Jamie L. Jensen.

**Investigation:** Rebecca L. Sansom, Jennifer B. Nielson, R. Steven Turley, Richard E. West, Geoffrey Wright, Jamie L. Jensen.

**Methodology:** Rebecca L. Sansom, Jennifer B. Nielson, R. Steven Turley, Richard E. West, Geoffrey Wright, Jamie L. Jensen.

**Project administration:** Rebecca L. Sansom, Jennifer B. Nielson, Jamie L. Jensen.

**Writing – original draft:** Jeffrey Shipley, Haley Mickelsen, Bryn St. Clair, Jamie L. Jensen.

**Writing – review & editing:** Bryn St. Clair.

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
