## [Decision Letter · Decision Letter 0]

20 Feb 2023

PONE-D-22-33194Iterating toward change: improving student-centered teaching through the STEM faculty institute (STEMFI)PLOS ONE

Dear Dr. Jensen,

Thank you for submitting your manuscript to PLOS ONE. After careful consideration, we feel that it has merit but does not fully meet PLOS ONE’s publication criteria as it currently stands. Therefore, we invite you to submit a revised version of the manuscript that addresses the points raised during the review process.

I expect you to submit your article for evaluation after reviewing and editing in terms of the issues highlighted by the reviewers.

We look forward to receiving your revised manuscript.

Kind regards,

Ayse Hilal Bati, Associate Professor

Academic Editor

PLOS ONE

Journal Requirements:

a) Did participants provide their written or verbal informed consent to participate in this study?

Additional Editor Comments :

Dear authors,

I expect you to submit your article for evaluation after reviewing and editing in terms of the issues highlighted by the reviewers.

Reviewers' comments:

Reviewer's Responses to Questions

**Comments to the Author**

1. Is the manuscript technically sound, and do the data support the conclusions?

Reviewer #1: Partly

Reviewer #2: Yes

Reviewer #3: Yes

2. Has the statistical analysis been performed appropriately and rigorously? 

Reviewer #1: N/A

Reviewer #2: Yes

Reviewer #3: Yes

3. Have the authors made all data underlying the findings in their manuscript fully available?

Reviewer #1: Yes

Reviewer #2: Yes

Reviewer #3: Yes

4. Is the manuscript presented in an intelligible fashion and written in standard English?

Reviewer #1: Yes

Reviewer #2: Yes

Reviewer #3: Yes

5. Review Comments to the Author

Reviewer #1: I have attached a word document with this information. I have two main concerns with this paper. 1. The classes selected to observe with COPUS at the end of the program were not randomly selected but rather the participants selected the courses to be observed. This introduces consider bias into the study design and undermines the results presented. 2. The results from the qualitative analysis did not further differentiate the participants in each of the categories the researchers presented and at times different arguments were presented for the same findings.

Reviewer #2: Overall, this is a much needed study in the field. However, there are some issues with the qualitative data that need to be addressed. The most major issue is that the qualitative data needs to revised to make it much more clear of the themes and the process of finding these themes. For example: how did the themes emerge? How prevalent are these themes? How were the themes found (i.e., using inductive or deductive approaches). Commentary and explanation of the provided quotes are also necessary in order to explain how the quotes are related to the theme. Additionally, the triangulation between the COPUS and interview data needs to be significantly more explained than it is now. Some visuals may be useful to represent the qualitative data.

Here is some line feedback I have as well:

Line 53: Often workshops are one-off professional development opportunities. You seem to be describing a course redesign institute here. Being clear about what you mean by workshops will be important.

Line 85: How framework is connected to teaching and learning is important to include.

Line 96-98: Additional explanation of how your description of the program is related to the attitude toward the behavior is important here.

Reviewer #3: The authors of this manuscript used Ajzen’s Theory of Planned Behavior to design the STEM Faculty Institute (STEMFI) and categorize the types of reformers who completed this professional development program. Pre and post surveys, interviews and classroom observations were used to create descriptive profiles of participants who changed their instructional styles following STEMFI. Based on the improvements of the majority of the participants, the researchers asserted that the Theory of Planned Behavior was an effective framework for producing change in faculty behavior.

Overall, I think that this manuscript was well-written with a sound rationale and sophisticated design, and analysis. I have no request for modifications. I felt inspired while reading this manuscript and I think that this content will be a substantial contribution to the study of faculty professional development programs.

6. PLOS authors have the option to publish the peer review history of their article (what does this mean?). If published, this will include your full peer review and any attached files.

Reviewer #1: No

Reviewer #2: **Yes: **Ashley Nicole Harlow

Reviewer #3: No

---

## [Author Response · Author response to Decision Letter 0]

4 Apr 2023

Response to Reviewers:

Reviewer #1: 

I have attached a word document with this information. I have two main concerns with this paper. 1. The classes selected to observe with COPUS at the end of the program were not randomly selected but rather the participants selected the courses to be observed. This introduces consider bias into the study design and undermines the results presented. 2. The results from the qualitative analysis did not further differentiate the participants in each of the categories the researchers presented and at times different arguments were presented for the same findings.

Major Concerns

#1. Including COPUS as an objective empirical assessment of participants actual teaching practices in the classroom had the potential of providing a rigorous assessment the program. However, the fact that the researcher allowed the STEMFI participants to select the classes to be COPUSed at the end of the program rather than randomly selecting classes as they did in the first semester, severely undermines the rigor of the results. By allowing the faculty to select the post-classses to COPUS, they introduced considerable bias into their results. Given that this study is now concluded, I am not sure how to resolve this issue.

Thank you for this feedback. We agree that allowing participants to choose the class post-workshop would be biased if we were trying to show that the instructor was truly reformed in all aspects of their teaching. However, we were just trying to show that instructors could take what they had learned and successfully implement it in their courses (hence, we wanted to see those particular courses). Given this distinction, however, we agree that our language throughout was perhaps not so clear on this point. We have gone through the paper and cleaned that up. We have also made a note of this in our limitations section.

#2. Combining both a quantitative and qualitative analysis was a nice idea. However, I did not see enough differentiation among the qualitative results for each categorization of faculty post involvement in the program to be helpful. It felt like all reform groups said the same thing yet the authors interpreted the results differently and went beyond the scope of the coding rubric they established and presented in the paper.

This is an interesting perspective and it made us re-think how we described our categories. We agree that it was difficult to differentiate among the reformer types with the qualitative data and we decided that perhaps that was not the point. We have gone in and rewritten the qualitative section to be more about the four themes that emerged in conjunction with the TPB framework, with suggestions for trends that match the profiles. But, we emphasize that it is not always a direct relationship between them (i.e., there is a lot of overlap). We have also included descriptions of our didactic non-reformers.

# 3. Under the Explanatory results-Descriptive profiles section of the paper you left out two key categories of participants: Interactive Reformers and Didactic Non-reformers. Neither of these groups changed over the course of the program. In many ways your qualitative data from these two groups would be as important or more important than the data from the groups who did change. 

See above - we have included additional descriptions of non-reformers.

Minor Concerns

Introduction

On line 43 the authors introduce the term student-centered teaching (STC). Could they please explain why they are using this term rather than active learning or evidence-based teaching practices. Using multiple terms for the same teaching methods can introduce confusion to the professional development field of study.

We prefer the term “Student-Centered Teaching” to emphasize the constructivist approach that we were teaching. Active learning seems too broad (i.e., teaching can be active and not constructivist). Evidence-based practices is also just not as descriptive. We have made sure we consistently use that term throughout.

Line 60. As the authors only cite one study [4], it would be best if this sentence began with the qualifier “some”. Also it is not clear what you mean by “including those with pedagogical training”. What does “those” refer to.

Thank you for catching this. We have changed this sentence.

Ln 63-67. A key citation that is missing from this section is “Henderson, C., Beach, A., & Finkelstein, N. (2011). Facilitating change in undergraduate STEM instructional practices: an analytic review of the literature. Journal of Research in Science Teaching, 48(8), 952–984. https:// doi. org/ 10.1002/ tea. 20439” Henderson et al. review of 191 articles describing professional development, very succinctly concludes that are two common practices that do not work and he identifies four practices of successful programs. STEMFI includes all of these.

 Thank you for the excellent paper and suggestion. We have added this.

Ln 70. An additional reference by Ann Austin would be most appropriate here as well.

Austin, A. E. (2011). Promoting evidence-based change in undergraduate science education. In Fourth committee meeting on status, contributions, and future directions of discipline-based education research.

Another useful citation. Thank you..

There is a good deal of repeated material in the Introduction and Methods section of the paper. I realize that the authors are try to align different components of STEMFI with their theoretical framework but it gets confusing. For instance, the information on Ln 77-81 is repeated in Methods and really does not fit in the introduction.

We have removed that paragraph.

This issue arises again in Ln 115-128. Most of this content in repeated in the Methods section starting at LN 149. Ln 125-128 particularly is methods rather than introduction.

Possibly the authors could move and consolidate the description “STEMFI Program” to the Introduction and address how you collected COPUS data and coded interview data in Methods.

We have removed redundancy in the Introduction and preferred to keep all methodologies in the methods.

Methods

Please present aggregate information on the gender and ethnicity of the participants.

We have included gender. Ethnicity data was not collected.

Please state how the four random class sessions were selected. Was COPUS conducted in person or by watching a video of class. If in person, was the faculty member aware of that this class was being observed? For the Post-COPUS observation, was this done at the end of the second term that they taught the same course or at the end of the first term they taught the course?

 We have further clarified this in the methods.

How long was the summer workshop and were faculty financially compensated for attendance.

We have added this to the methods.

Please indicate who served as peer mentors and how were mentors paired with determined.

 We have added this, as well

One major concern is your post-COPUS observation method. You state that you allowed the faculty to select the post-course to be observed. In the paper you gave your explanation for this (ln 185-187). However, I see this as a major limitation of your study. It is possible that the faculty member only used the new teaching method on those 3-4 days and used didactic teaching methods on the other 40 days of teaching. Therefore, did they really change and was your categorization of them (fig 2) correct. The post-COPUS score would indicate how well they implemented the select group or type of evidence-based teaching they used that day but may not reflect their actual teaching method. You need to put many more qualifiers on this part of your results.

Thank you. We have commented on this above and we have made additions to our paper to indicate this potential limitation.

The categories you created to classify participants are very clever and the figure 2 is a nice representation of that categorization method.

Thank you!

Interview Protocol

You state that your interview protocol was to align with the Theory of Planned Behavior (TPB) and address attitudes, student response and confidence. However, you then created four themes but there were only three attributes of the TPB. Two of your themes 1) attitude and 2 student responses align directly with TPB but motivation and challenges do not align. This was confusing. Could you explain how the TPB informed the last two themes of motivation and challenges. You may have to do this in results section.

Thank you for asking for this clarification. In our rewrite of our qualitative data results section, we have better clarified how these fit. It’s important to note that the interviews were only semi-structured and the data were analyzed using an emergent themes approach. Thus, two themes that naturally emerged went beyond TPB. We have included this in our methods.

The sentences from 224 to 227 do not fit under Interview Protocol but rather should be a stand alone paragraph indicating how you will triangulate your quantitative and qualitative data.

 Done

Results

Quantitative

Please add the academic rank to the participants in each category. You could add three rows under your existing row at bottom of Fig 2 and indicate the number of faculty who are Assist., Assoc., Full Professors. This would help the reader see if newer faculty are more likely to change than established faculty.

We felt that adding it to the figure was just a little too messy. But we have included that information in the text of the results.

Qualitative results

The sentence on Ln 249 starting with “Attitudes… should be the beginning of a new paragraph as you are now presenting results on your first theme of attitudes.

Okay.

The quotes you presented nicely support the binary categories you established.

Thank you!

Explanatory Results- Descriptive profiles

This section was challenging to follow. I found myself creating a chart with the four themes in the first column and each column was one of your faculty categories, i.e. Beginning Reformer, Dramatic Reformer, etc. and putting the binary classification in each appropriate cell. I strongly suggest you create such a table.

Also, rather than just presenting the results from the coding of the interviews there is a good deal of interpretation of these results. Keeping the interpretation of results for the discussion section could be helpful to the reader and allow more compare and contrasting between faculty categories.

I am also concerned that the results for this section do not add further clarity to your results. When I look at the results for motivation all four faculty categories show intrinsic motivation and all four show logistical challenges. On Ln 430, you indicate that student centered reformers perceived the challenge differently but what method did you use to code perception? The authors may be reading too much into faculty answers.

When you presented results for Advanced Reformer as well as Student-centered Reformer, you changed the order of presentation of the four themes. Please keep the same order as the others, i.e. attitude, student response, motivation and then challenges. This section has high cognitive load for the reader and keeping the same order of presentation of themes within each category would help decrease the load.

I was very much looking forward to your qualitative triangulation of the Interactive and Didactic Reformer. What did you find in your interviews with them that could shed light on faculty who do not change. These are really the major challenge for future professional development programs. It is necessary that you provide the explanatory results for these two profiles.

We agree with you completely. We have significantly changed this section to try to make it clearer and more useful for interpretation. We have also included descriptions for non-reformers.

Discussion

As I mentioned earlier, the interpretation of result found in the results section would fit better in the discussion section.

Agreed. We have tried to move our interpretations all to the discussion.

I did not see data presented to support your claim in Ln 455 “By addressing the attitudes….”

We have modified this sentence to be more clear.

Ln. 471. The sentence starting with “our results…” needs a qualifier as not all faculty changed categories by the end of the program. I would suggest “SCT was instrumental in aiding 80% (or the majority of participants) STEMFI participants. 

We have modified this sentence.

Ln 473. I am not clear how STEMFI program specifically addressed the participants attitudes.

We have clarified the claim in this paragraph.

Reviewer #2: Overall, this is a much needed study in the field. However, there are some issues with the qualitative data that need to be addressed. The most major issue is that the qualitative data needs to revised to make it much more clear of the themes and the process of finding these themes. For example: how did the themes emerge? How prevalent are these themes? How were the themes found (i.e., using inductive or deductive approaches). Commentary and explanation of the provided quotes are also necessary in order to explain how the quotes are related to the theme. Additionally, the triangulation between the COPUS and interview data needs to be significantly more explained than it is now. Some visuals may be useful to represent the qualitative data.

Thank you for the feedback. We have added a little bit more detail to methods explaining the thematic analysis. We did not, however, quantify themes as that was not the focus of the analysis. We do agree that the profiles portion of our quantitative analysis (i.e., the triangulation between copus “reformer types” and interview data) was not as clear as we had hoped (as another reviewer pointed out). We have opted to focus on the four themes and their relation to the Theory of Planned Behavior. And we have explained in our discussion how the qualitative data was not able to differentiate “reformer types” like the quantitative could.

Here is some line feedback I have as well:

Line 53: Often workshops are one-off professional development opportunities. You seem to be describing a course redesign institute here. Being clear about what you mean by workshops will be important. 

This was a good point. We have added additional detail to clearly show that this program is much more involved than a one-off PD experience.

Line 85: How framework is connected to teaching and learning is important to include.

Great point. We have added additional detail to that paragraph.

Line 96-98: Additional explanation of how your description of the program is related to the attitude toward the behavior is important here.

Thank you. We have clarified.

Reviewer #3: The authors of this manuscript used Ajzen’s Theory of Planned Behavior to design the STEM Faculty Institute (STEMFI) and categorize the types of reformers who completed this professional development program. Pre and post surveys, interviews and classroom observations were used to create descriptive profiles of participants who changed their instructional styles following STEMFI. Based on the improvements of the majority of the participants, the researchers asserted that the Theory of Planned Behavior was an effective framework for producing change in faculty behavior.

Overall, I think that this manuscript was well-written with a sound rationale and sophisticated design, and analysis. I have no request for modifications. I felt inspired while reading this manuscript and I think that this content will be a substantial contribution to the study of faculty professional development programs.

Thank you so much!

---

## [Decision Letter · Decision Letter 1]

22 May 2023

PONE-D-22-33194R1Iterating toward change: improving student-centered teaching through the STEM faculty institute (STEMFI)PLOS ONE

Dear Dr. Jensen,

Thank you for submitting your manuscript to PLOS ONE. After careful consideration, we feel that it has merit but does not fully meet PLOS ONE’s publication criteria as it currently stands. Therefore, we invite you to submit a revised version of the manuscript that addresses the points raised during the review process.

Dear Author/s
Your article should be edited in line with reviewer suggestions.
It will be evaluated by me after this arrangement.

We look forward to receiving your revised manuscript.

Kind regards,

Ayse Hilal Bati, Professor

Academic Editor

PLOS ONE

Journal Requirements:

Additional Editor Comments:

Dear Author/s

Your article should be edited in line with reviewer suggestions. It will be evaluated by me after this arrangement.

Reviewers' comments:

Reviewer's Responses to Questions

**Comments to the Author**

1. If the authors have adequately addressed your comments raised in a previous round of review and you feel that this manuscript is now acceptable for publication, you may indicate that here to bypass the “Comments to the Author” section, enter your conflict of interest statement in the “Confidential to Editor” section, and submit your "Accept" recommendation.

Reviewer #1: All comments have been addressed

Reviewer #2: (No Response)

Reviewer #3: All comments have been addressed

Reviewer #4: All comments have been addressed

Reviewer #5: (No Response)

Reviewer #6: All comments have been addressed

Reviewer #7: (No Response)

Reviewer #8: (No Response)

2. Is the manuscript technically sound, and do the data support the conclusions?

Reviewer #1: Yes

Reviewer #2: Partly

Reviewer #3: Yes

Reviewer #4: Yes

Reviewer #5: Partly

Reviewer #6: Yes

Reviewer #7: Partly

Reviewer #8: Partly

3. Has the statistical analysis been performed appropriately and rigorously? 

Reviewer #1: Yes

Reviewer #2: Yes

Reviewer #3: Yes

Reviewer #4: Yes

Reviewer #5: N/A

Reviewer #6: Yes

Reviewer #7: N/A

Reviewer #8: N/A

4. Have the authors made all data underlying the findings in their manuscript fully available?

Reviewer #1: Yes

Reviewer #2: Yes

Reviewer #3: Yes

Reviewer #4: No

Reviewer #5: (No Response)

Reviewer #6: Yes

Reviewer #7: Yes

Reviewer #8: No

5. Is the manuscript presented in an intelligible fashion and written in standard English?

Reviewer #1: Yes

Reviewer #2: Yes

Reviewer #3: Yes

Reviewer #4: Yes

Reviewer #5: Yes

Reviewer #6: Yes

Reviewer #7: No

Reviewer #8: Yes

6. Review Comments to the Author

Reviewer #1: May 6, 2023

Re-Review PLoS One PONE-D-22-33194

The authors have done a very nice job of revising their manuscript based on my initial comments. I find the manuscript is now easier to read and presents a stronger case for their findings. I have only a few minor changes and I do not need to rereview these minor changes.

Unfortunately the manuscript did not come with line numbers so I will only be able to use page number to indicate areas for change.

Page 7

You start off this section indicating that the program consisted of three phases: pre-, during, and post-workshop. You indicate the Pre-workshop content and in paragraph 2 on page 8 you identify phase three. Where is Phase 2. It would be nice if paragraph one on page 8 began with the phrase, Phase 2 involved faculty participation in a on-time workshop.

On the other hand you could just drop the use of the term “phases:.

Page 9 top

“observed using COPUS to measure their post teaching behaviors. Participants chose which classes would be observed.”

Please remove the sentence “Participants chose which classes would be observed” as you more fully explain this in the next section about Observations. This much detail feels out of place here.

Page 9 bottom

“Post-observations were made in the first semester they taught following their participation and were requested by the participant in order to showcase the new techniques they were planning to use.”

This is a nice sentence that addresses the non-random selection of post-workshop courses to COPUS.

The word “selected” would be a better word than “requested”

Page 10

“likewise those who were already using student-centered strategies and continued

being student-centered were labeled “Student-Centered Reformers”.

I found it odd that a faculty member that started as student-Centered and remained at student -centered would be titles a “reformer” as in fact they did not reform they just kept doing what they were already doing. If however they “broadened their strategies’ as was stated for the interactive to interactive faculty than I could accept labeling them as reformers. Please add more text here to show how they can be considered “reformers”.

This will also impact your statement on page 12 that states 35 of 41 participants changed their teaching enough to be “reformers”. If faculty who remained in their category ( Inter to Inter, or SC to SC) and did not add new strategies, they should not be termed reformers.

Page 13-top of page.

“so they can only hint at potential differences between reforming attitudes.”

I would rewrite as “so they can only hint at potential differences between reformers.”

Page 13

You italicized the term in transition but did not italicize the dichotomus pair “Fully Reformed”. As you do not italicize the dichotomous pairs under the other themes please remove the italics from in transition.

Page 21

Student-centered reformers were those who already had demonstrable experience implementing SCT strategies coming into this experience. Their attitudes can teach us several

things. First, we see fully reformed attitudes among this group

The middle sentence, Their attitudes.. needs a rewrite. I suggest the following

Student-centered reformers were those who already had demonstrable experience implementing SCT strategies coming into this experience. We found that this groups has fully reformed attitudes.

Page 21- bottom

Non-reformers displayed many of the same attitudes as those who chose to make

measurable changes. However, we can learn a few things from them that can help inform future efforts.

The second sentence felt a bit too casual with your reader. I would just remove it.

Page 24 Conclusions

“According to the established literature, students who learn through a more student-centered

approach have a greater overall retention of information [e.g., 31].”

I have read citation 31 Smith et al. 2014. They conducted a survey of teaching practices across their university using COPUS and had faculty self-report the practices they use. They noted a high degree of alignment between COPUS results and self-reports. I did not see any mention of “greater overall retention of information”. Please remove this citation.

You may need to remove this whole last sentence because I am unaware of any established literature that has shown greater overall retention of information. There are many citations that support increased exam scores but none that I know of that deal with retention.

Reviewer #2: Overall, my biggest feedback is what Reviewer 1 previously mentioned about the post-observation not being at random and only occurring the single time when the pre-observation occurred multiple times. This is problematic as the instructor could be performative in their methods and not actually adopting SCT strategies. Additionally, only doing the post-observation once, is a similar issue that leads for the data to not be as generalizable on if the class was a one-off or if the instructor has really adopted SCT strategies. Overall, I think to help remediate this, the categorization of faculty needs to be written differently as you can’t confirm that they are now student-centered instructors, but rather that they can use the techniques efficiently. This needs to be made clear in the manuscript.

Reviewer #3: I appreciate the edits made to further clarify the results and methods sections of this manuscript. The faculty professional development community will greatly benefit from the review of this study and program.

Reviewer #4: This submission is interesting, well written and an important topic. The authors discuss the theoretical concepts leading to their intervention, which adds to the depth of the submission. The intervention and it's evaluation are also multi-layered and the authors have already made some helpful changes to assist the reader in understanding their results e.g. Figure 2. I would like to request some minor changes. 1. Please describe acronyms fully when first introduced. The paper assumes prior knowledge e.g. STEM. COPUS is described but only on Page 7 when the acronym was introduced much earlier. 2. The link provided to the data repository didn't list the study https://scholarsarchive.byu.edu/ - please provide full details about where the data is located. 3. The authors discuss the concept in the introduction that "development programs frequently do not cause lasting changes to teaching strategies or student engagement" - in the discussion please address how you plan to overcome this issue. I wouldn't classify the current duration of observation as 'long-term'.

Reviewer #5: This study evaluated the effectiveness of a STEMFI on student-centered teaching. Both classroom observations with COPUS and post-interviews were used for evaluation. The results showed that faculty shifted toward student-centered teaching after the STEMFI program. The manuscript is well-written, and the findings are interesting. However, some important clarifications about the results are needed. 1. COPUS captures teacher and student behaviors (e.g., lecturing, asking questions, group work), but it does not capture the specific student-centered strategies (e.g., think-pair-share, 5E learning cycle). Faculty shifted toward student-centered teaching could be because they spent more time on group work, but does not necessarily mean they used more student-centered strategies. Also, the characteristics of each COPUS profile (in terms of the COPUS codes) need to be discussed to help readers interpret the results, especially those who are not familiar with COPUS. Please clarify for each type of reformer, what student-centered strategies they tried after participation, or they used the same strategies as before but spent more class time on those strategies. 2. I am concerned about the reliability of the observation data. How were graduate and undergraduate students trained for using COPUS? Have you investigated the inter-rater reliability?

More detailed comments below:

1. In my opinion, COPUS codes and the characteristics of each COPUS profile should be discussed in more detail in order to help readers interpret the results. COPUS codes are somewhat general, which doesn’t capture specific student-centered strategies. I think the authors need to be more carefully describing how faculty reformed their teaching. Did they spend less time lecturing but more time on group work, or did they implement some new student-centered strategies, such as 5E learning cycle and POGIL? Also, I think you also need to give more details about the strategies introduced in STEMFI. For example, what is POGIL if someone doesn’t already know.

2. It is unclear to me how graduate and undergraduate students were trained for using COPUS. How many students did the observations? How were they assigned for observations for different faculty? What is the inter-rater reliability?

3. On page 12, the authors said some post-data was collected on hybrid or online courses. How did it affect the classroom observations? I imagine it is hard to do COPUS with online courses. Also, how did it affect the COPUS profile? Were online courses more likely to be didactic?

4. Thematic analysis. How many researchers coded the transcripts? Did you always have two researchers do independent coding first?

5. Please give more details about the triangulation of COPUS and interview results. Please say more about the explanatory mixed methods design, and how you “merge” the findings.

6. Did the faculty need to apply to get into the program? If so, how many applications each year? Also, I think those who applied are the ones more interested in SCT, and may have more positive attitudes toward SCT. Did you do pre-interview? Any chance those participants already had positive attitudes toward SCT before the program? Can you please comment on how this can affect your findings?

7. You have student responses and challenges as two separate themes. The literature has shown that students’ negative responses is one barrier for reformed teaching. Can you please comment on why you decided not to include this as a challenge? On page 18, there is a quote of student negative feedback as a perceived challenge.

8. The development of the interview protocol was informed by TPB. Am I understanding it correctly that the students were considered “significant others” and students’ responses gave subject norms? If you could be more explicitly on how each of the three factors informed the interview questions, that would be great.

9. You included faculty ranking when reporting the results. Did you see any patterns of faculty at different rankings?

Thank you for your work! Looking forward to your responses.

Reviewer #6: An excellent addition to the literature on faculty teaching development. The authors have responded well to the comments from initial review and made updates that improve the quality and readability of their manuscript. I have no further feedback to offer, and look forward to seeing this manuscript published!

Reviewer #7: There are minor revisions (e.g., use of terminology, condensing the Discussion). Careful edit of the text is necessary.

Reviewer #8: Summary:

This study attempts to examine the impacts of a faculty professional development program, STEMFI, on the impacts on instructional practices (COPUS data) and instructor’s attitudes, perceptions of students’ responses, and confidence (interview data). The study appears to be well aligned to the selected theoretical framework, The Theory of Planned Behavior. And there are some clear visual representations in Figures 1 and 2. Unfortunately, there are quite a few major revisions that need to be addressed before this manuscript is considered any further: (1) focus is on examining student-centered teaching (SCT), when I think it should be on active learning; (2) educational problem(s) and/or research question(s) are absent (or at least unclear); (3) COPUS data are presented in a qualitative manner, but authors claim that these are quantitative results; and (4) mixed methods research design is unclear. I suggest that the authors rework the introduction and methods with the suggestions below in mind.

Major issues:

Freeman et al. (2014) meta-analysis focuses on the impacts of active learning on student performance outcomes, not student-centered teaching (SCT). I would be careful to not conflate active learning and SCT. I would suggest that the terminology is changed from SCT to active learning since not all active learning is necessarily student-centered.

Relatedly, is there a reason that you decided to write your own definition of SCT (or what I am suggesting is active learning in my point above)? Why did you not want to use a definition from the literature? I suggest that you review this CBE LSE paper on active learning for potential literature-based definitions: Driessen, E. P., Knight, J. K., Smith, M. K., & Ballen, C. J. (2020). Demystifying the meaning of active learning in postsecondary biology education. CBE—Life Sciences Education, 19(4), ar52.

The educational problem(s) and/or research question(s) are not clearly stated in the abstract nor at the end of the introduction. I am not sure what is being examined in this study, so it will be hard for me to determine if the methods are appropriate and well-aligned with the educational problem and/or research question.

While Figure 2 is clear for understanding how groups of instructors shifted from one COPUS profile to another, I don’t think that Figure 2 is sufficient for the reader to be able to understand the quantitative nature of the COPUS results. Right now, the results are described in a qualitative manner, which is okay; however, the authors claim that the results are quantitative. I would suggest that you give an example of the instructor and student behaviors occurring in each of those six types of reformer classrooms. See Shi et al. (2023) CBE LSE (https://doi.org/10.1187/cbe.22-03-0047) for an example of figures.

Also, the mixed methods research design is unclear to me. Right now, it appears that COPUS data were collected, and interview data were collected, but how one dataset might inform the other needs to be further explored by the authors to answer the research question(s).

Minor issues:

Introduction:

Page 3

Please add a reference to the last statement on this page about the primary objective of these workshop. In addition, I think that “knowledge” is missing from the list of primary outcomes of faculty professional development programs.

Methods:

Page 7

Participants: Did you collect any other demographics data from the instructors? Or only discipline and gender (in the binary)? Also, could you please describe the student population being served by these instructors being studied? And finally, please described the process for recruiting these instructors.

Page 9

How did you ensure that the students were trained to code COPUS in a reliable manner? Did you calculate inter-rater reliability after this training? Please provide more details on how you tested for coder reliability.

Page 10

I was not able to see the supplemental information to be able to see the full interview protocol. Also, it would be helpful if the supplemental file information (e.g., Supplemental File S1) was noted in the text.

Page 11

Who are the several researchers that thematically coded the interview responses? I think it is important to be transparent about which specific researchers did this work to build trustworthiness in the data.

How exactly did you connect the quantitative and qualitative data using exploratory mixed methods design? It is unclear to me which COPUS and which interview data were used for these analyses. It is not until the results that I read that themes were not tied to specific reformer types. It is important that you bring up these details in the methods, not just results.

Results:

Page 11

I would move this participant info to the methods section. It’s good to be transparent about the loss of participants due to COVID, but I don’t think it is a great way to start your results section.

Why did you decide to add the ranking of the instructors to the quantitative results? Are you interested in examining how instructor rankings impacted use of student-centered teaching practices? Again, I am not sure if these results are aligned to the research questions and methods as I did not clearly see research questions earlier on.

References:

Check first author spelling for this citation: Creswell, J. W., & Plano Clark, V. L. (2007). Designing and conducting mixed methods research. Thousand Oaks, CA: Sage.

7. PLOS authors have the option to publish the peer review history of their article (what does this mean?). If published, this will include your full peer review and any attached files.

Reviewer #1: No

Reviewer #2: No

Reviewer #3: No

Reviewer #4: **Yes: **Richard G McGee

Reviewer #5: No

Reviewer #6: No

Reviewer #7: No

Reviewer #8: No

---

## [Author Response · Author response to Decision Letter 1]

30 May 2023

Response to Reviewers

Reviewer #1: May 6, 2023

Re-Review PLoS One PONE-D-22-33194

The authors have done a very nice job of revising their manuscript based on my initial comments. I find the manuscript is now easier to read and presents a stronger case for their findings. I have only a few minor changes and I do not need to rereview these minor changes.

Unfortunately the manuscript did not come with line numbers so I will only be able to use page number to indicate areas for change.

Page 7

You start off this section indicating that the program consisted of three phases: pre-, during, and post-workshop. You indicate the Pre-workshop content and in paragraph 2 on page 8 you identify phase three. Where is Phase 2. It would be nice if paragraph one on page 8 began with the phrase, Phase 2 involved faculty participation in a on-time workshop.

On the other hand you could just drop the use of the term “phases:.

Good suggestion. We have added this phrase for clarification.

Page 9 top

“observed using COPUS to measure their post teaching behaviors. Participants chose which classes would be observed.”

Please remove the sentence “Participants chose which classes would be observed” as you more fully explain this in the next section about Observations. This much detail feels out of place here.

Good suggestion. We have removed this sentence for clarification.

Page 9 bottom

“Post-observations were made in the first semester they taught following their participation and were requested by the participant in order to showcase the new techniques they were planning to use.”

This is a nice sentence that addresses the non-random selection of post-workshop courses to COPUS.

The word “selected” would be a better word than “requested”

Thank you for this suggestion. We have adjusted the wording.

Page 10

“likewise those who were already using student-centered strategies and continued

being student-centered were labeled “Student-Centered Reformers”.

I found it odd that a faculty member that started as student-Centered and remained at student -centered would be titles a “reformer” as in fact they did not reform they just kept doing what they were already doing. If however they “broadened their strategies’ as was stated for the interactive to interactive faculty than I could accept labeling them as reformers. Please add more text here to show how they can be considered “reformers”.

This will also impact your statement on page 12 that states 35 of 41 participants changed their teaching enough to be “reformers”. If faculty who remained in their category ( Inter to Inter, or SC to SC) and did not add new strategies, they should not be termed reformers.

We have added more text to page 10 and 12 to clarify the description of classification.

Page 13-top of page.

“so they can only hint at potential differences between reforming attitudes.”

I would rewrite as “so they can only hint at potential differences between reformers.”

Thank you, we adjusted this wording.

Page 13

You italicized the term in transition but did not italicize the dichotomus pair “Fully Reformed”. As you do not italicize the dichotomous pairs under the other themes please remove the italics from in transition.

Adjusted.

Page 21

Student-centered reformers were those who already had demonstrable experience implementing SCT strategies coming into this experience. Their attitudes can teach us several

things. First, we see fully reformed attitudes among this group

The middle sentence, Their attitudes.. needs a rewrite. I suggest the following

Student-centered reformers were those who already had demonstrable experience implementing SCT strategies coming into this experience. We found that this groups has fully reformed attitudes.

Great suggestion. We have implemented the edit.

Page 21- bottom

Non-reformers displayed many of the same attitudes as those who chose to make

measurable changes. However, we can learn a few things from them that can help inform future efforts.

The second sentence felt a bit too casual with your reader. I would just remove it.

We have removed colloquial wording.

Page 24 Conclusions

“According to the established literature, students who learn through a more student-centered

approach have a greater overall retention of information [e.g., 31].”

I have read citation 31 Smith et al. 2014. They conducted a survey of teaching practices across their university using COPUS and had faculty self-report the practices they use. They noted a high degree of alignment between COPUS results and self-reports. I did not see any mention of “greater overall retention of information”. Please remove this citation.

You may need to remove this whole last sentence because I am unaware of any established literature that has shown greater overall retention of information. There are many citations that support increased exam scores but none that I know of that deal with retention.

This is an excellent point. We have removed the sentence entirely.

Reviewer #2: Overall, my biggest feedback is what Reviewer 1 previously mentioned about the post-observation not being at random and only occurring the single time when the pre-observation occurred multiple times. This is problematic as the instructor could be performative in their methods and not actually adopting SCT strategies. Additionally, only doing the post-observation once, is a similar issue that leads for the data to not be as generalizable on if the class was a one-off or if the instructor has really adopted SCT strategies. Overall, I think to help remediate this, the categorization of faculty needs to be written differently as you can’t confirm that they are now student-centered instructors, but rather that they can use the techniques efficiently. This needs to be made clear in the manuscript.

Throughout the manuscript we have added and softened the claims that the participants became student-centered instructors, but rather that they can use the techniques now more efficiently.

Reviewer #3: I appreciate the edits made to further clarify the results and methods sections of this manuscript. The faculty professional development community will greatly benefit from the review of this study and program.

Thank you for your time in reviewing and encouragement. We are excited to share our results.

Reviewer #4: This submission is interesting, well written and an important topic. The authors discuss the theoretical concepts leading to their intervention, which adds to the depth of the submission. The intervention and it's evaluation are also multi-layered and the authors have already made some helpful changes to assist the reader in understanding their results e.g. Figure 2. I would like to request some minor changes. 

1. Please describe acronyms fully when first introduced. The paper assumes prior knowledge e.g. STEM. COPUS is described but only on Page 7 when the acronym was introduced much earlier. 

We have clarified the acronyms. Thank you for pointing out this edit.

2. The link provided to the data repository didn't list the study https://scholarsarchive.byu.edu/ - please provide full details about where the data is located.

Thank you for the reminder. We were waiting to post it, until the manuscript was accepted. We have fixed the URL to access the data that is now posted: https://scholarsarchive.byu.edu/data/49

 3. The authors discuss the concept in the introduction that "development programs frequently do not cause lasting changes to teaching strategies or student engagement" - in the discussion please address how you plan to overcome this issue. I wouldn't classify the current duration of observation as 'long-term'.

In the discussion we have qualified our perspectives on lasting change based on the evidence from this study. Thank you for helping us connect these ideas in our paper.

Reviewer #5: This study evaluated the effectiveness of a STEMFI on student-centered teaching. Both classroom observations with COPUS and post-interviews were used for evaluation. The results showed that faculty shifted toward student-centered teaching after the STEMFI program. The manuscript is well-written, and the findings are interesting. However, some important clarifications about the results are needed. 1. COPUS captures teacher and student behaviors (e.g., lecturing, asking questions, group work), but it does not capture the specific student-centered strategies (e.g., think-pair-share, 5E learning cycle). Faculty shifted toward student-centered teaching could be because they spent more time on group work, but does not necessarily mean they used more student-centered strategies. Also, the characteristics of each COPUS profile (in terms of the COPUS codes) need to be discussed to help readers interpret the results, especially those who are not familiar with COPUS. Please clarify for each type of reformer, what student-centered strategies they tried after participation, or they used the same strategies as before but spent more class time on those strategies. 2. I am concerned about the reliability of the observation data. How were graduate and undergraduate students trained for using COPUS? Have you investigated the inter-rater reliability?

We have responded to each of the detailed comments below.

More detailed comments below:

1. In my opinion, COPUS codes and the characteristics of each COPUS profile should be discussed in more detail in order to help readers interpret the results. COPUS codes are somewhat general, which doesn’t capture specific student-centered strategies. I think the authors need to be more carefully describing how faculty reformed their teaching. Did they spend less time lecturing but more time on group work, or did they implement some new student-centered strategies, such as 5E learning cycle and POGIL? Also, I think you also need to give more details about the strategies introduced in STEMFI. For example, what is POGIL if someone doesn’t already know.

The COPUS measurement allows us to distinguish active learning strategies into a few categories. Codes include more teacher-centered approaches, such as lecturing by the instructor and listening by the students, more interactive strategies like student questions and instructor answers, and more fully student-centered strategies like group work, clicker questions, and the instructor moving and guiding around the room. It does not, however, assess specific active techniques. For more description of the instrument and codes, see Smith et al 2018. We have clarified the manuscript to include this description. Thank you for pointing out this edit. Additionally, we have included brief detail of the workshop but referred readers to our other publication where the workshop is described in more detail.

We have added information and an additional citation about the specific strategies introduced in STEMFI. 

2. It is unclear to me how graduate and undergraduate students were trained for using COPUS. How many students did the observations? How were they assigned for observations for different faculty? What is the inter-rater reliability?

We used the training protocol outlined in Smith et al. This is already included in the manuscript. This is a common training protocol used for the COPUS. (“ Undergraduate and graduate students were trained to use the COPUS using the training protocol established in Smith et al. [21].”)

3. On page 12, the authors said some post-data was collected on hybrid or online courses. How did it affect the classroom observations? I imagine it is hard to do COPUS with online courses. Also, how did it affect the COPUS profile? Were online courses more likely to be didactic?

We have added a clarifying statement that addresses the data collected on hybrid courses and how it affected the classroom observations.

4. Thematic analysis. How many researchers coded the transcripts? Did you always have two researchers do independent coding first?

Please see the description that was already revised from previous reviewers under the heading “Interview protocol for qualitative analysis”

5. Please give more details about the triangulation of COPUS and interview results. Please say more about the explanatory mixed methods design, and how you “merge” the findings.

Previous reviewers suggested that the triangulation was not clear. We significantly re-wrote the qualitative portion, responding to reviewers as such:

“This is an interesting perspective and it made us re-think how we described our categories. We agree that it was difficult to differentiate among the reformer types with the qualitative data and we decided that perhaps that was not the point. We have gone in and rewritten the qualitative section to be more about the four themes that emerged in conjunction with the TPB framework, with suggestions for trends that match the profiles. But, we emphasize that it is not always a direct relationship between them (i.e., there is a lot of overlap). We have also included descriptions of our didactic non-reformers.”

We also added to our manuscript, the following: “Through the lens of the Theory of Planned Behavior, we triangulated these categories with COPUS data to make loose hypotheses about the motivations of each reformer. However, we found that the themes were not robustly tied to specific reformer types in an exclusive way, so they can only hint at potential differences between reformers.”

Without more details about what you’d like to see, it is difficult to know what additional information is needed.

6. Did the faculty need to apply to get into the program? If so, how many applications each year? Also, I think those who applied are the ones more interested in SCT, and may have more positive attitudes toward SCT. Did you do pre-interview? Any chance those participants already had positive attitudes toward SCT before the program? Can you please comment on how this can affect your findings?

Thank you for this suggestion. We have included a clarification and future research direction in our limitation section about our sample that may have already had an increased interest in reforming their instruction.

7. You have student responses and challenges as two separate themes. The literature has shown that students’ negative responses is one barrier for reformed teaching. Can you please comment on why you decided not to include this as a challenge? On page 18, there is a quote of student negative feedback as a perceived challenge.

We recognize that negative student responses are a challenge (we added an additional sentence to clarify), but we separated this as a different category because there were also positive student responses. This is an important idea, however and we see this as a future avenue for specific and directed research. We have added a clarifying statement in the manuscript.

8. The development of the interview protocol was informed by TPB. Am I understanding it correctly that the students were considered “significant others” and students’ responses gave subject norms? If you could be more explicitly on how each of the three factors informed the interview questions, that would be great.

Thank you for this clarifying point. Yes, you are correct, students were considered the “significant others” that were providing feedback. But, we also considered peers and administration in our interviews. We have added details under the interview protocol for qualitative analysis to demonstrate how we developed the interview questions based on TPB.

9. You included faculty ranking when reporting the results. Did you see any patterns of faculty at different rankings?

In this iteration we didn’t have sufficient participation at each rank to see any meaningful patterns. Additionally, rank data was added at the request of a previous reviewer. We agree that this would be an important idea to flush out in future research questions.

Thank you for your work! Looking forward to your responses.

Reviewer #6: An excellent addition to the literature on faculty teaching development. The authors have responded well to the comments from initial review and made updates that improve the quality and readability of their manuscript. I have no further feedback to offer, and look forward to seeing this manuscript published!

Thank you for the feedback.

Reviewer #7: There are minor revisions (e.g., use of terminology, condensing the Discussion). Careful edit of the text is necessary.

We are unable to address the concerns of this reviewer since no details were offered. We have read through the manuscript to look for any grammatical errors and fixed any we found.

Reviewer #8: Summary:

This study attempts to examine the impacts of a faculty professional development program, STEMFI, on the impacts on instructional practices (COPUS data) and instructor’s attitudes, perceptions of students’ responses, and confidence (interview data). The study appears to be well aligned to the selected theoretical framework, The Theory of Planned Behavior. And there are some clear visual representations in Figures 1 and 2. Unfortunately, there are quite a few major revisions that need to be addressed before this manuscript is considered any further: (1) focus is on examining student-centered teaching (SCT), when I think it should be on active learning; (2) educational problem(s) and/or research question(s) are absent (or at least unclear); (3) COPUS data are presented in a qualitative manner, but authors claim that these are quantitative results; and (4) mixed methods research design is unclear. I suggest that the authors rework the introduction and methods with the suggestions below in mind.

Major issues:

Freeman et al. (2014) meta-analysis focuses on the impacts of active learning on student performance outcomes, not student-centered teaching (SCT). I would be careful to not conflate active learning and SCT. I would suggest that the terminology is changed from SCT to active learning since not all active learning is necessarily student-centered.

We have clarified in the manuscript on the STEMFI focus upon student-centered strategies. And we have modified the manuscript in places where we are just referring to active learning and places where we are specifically focused on SCT. We do not want to rename SCT throughout because our STEMFI program was specifically focused on SCT.

Relatedly, is there a reason that you decided to write your own definition of SCT (or what I am suggesting is active learning in my point above)? Why did you not want to use a definition from the literature? I suggest that you review this CBE LSE paper on active learning for potential literature-based definitions: Driessen, E. P., Knight, J. K., Smith, M. K., & Ballen, C. J. (2020). Demystifying the meaning of active learning in postsecondary biology education. CBE—Life Sciences Education, 19(4), ar52.

Reviewer #1 in the original response to reviews brought this up and we decided that student-centered teaching was the better term to use. See our response: We prefer the term “Student-Centered Teaching” to emphasize the constructivist approach that we were teaching. Active learning seems too broad (i.e., teaching can be active and not constructivist). Evidence-based practices is also just not as descriptive. We have made sure we consistently use that term throughout.

The educational problem(s) and/or research question(s) are not clearly stated in the abstract nor at the end of the introduction. I am not sure what is being examined in this study, so it will be hard for me to determine if the methods are appropriate and well-aligned with the educational problem and/or research question.

We have clarified our research question at the end of the introduction. 

While Figure 2 is clear for understanding how groups of instructors shifted from one COPUS profile to another, I don’t think that Figure 2 is sufficient for the reader to be able to understand the quantitative nature of the COPUS results. Right now, the results are described in a qualitative manner, which is okay; however, the authors claim that the results are quantitative. I would suggest that you give an example of the instructor and student behaviors occurring in each of those six types of reformer classrooms. See Shi et al. (2023) CBE LSE (https://doi.org/10.1187/cbe.22-03-0047) for an example of figures.

We have cited Stains et al. in our methodology of classification. We have further described our classification of reformers based on the instructor and student behavior data. Additionally, we feel like Figure 2 is representative according to previous reviewer comments by reviewer #1: “The categories you created to classify participants are very clever and the figure 2 is a nice representation of that categorization method.”

Also, the mixed methods research design is unclear to me. Right now, it appears that COPUS data were collected, and interview data were collected, but how one dataset might inform the other needs to be further explored by the authors to answer the research question(s).

Please see our comment above.

Minor issues:

Introduction:

Page 3

Please add a reference to the last statement on this page about the primary objective of these workshop. In addition, I think that “knowledge” is missing from the list of primary outcomes of faculty professional development programs.

We removed this sentence. It seemed redundant. We agree that knowledge should be a desired outcome, but we did not find studies that measured this.

Methods:

Page 7

Participants: Did you collect any other demographics data from the instructors? Or only discipline and gender (in the binary)? Also, could you please describe the student population being served by these instructors being studied? And finally, please describe the process for recruiting these instructors.

No, we did not collect any additional demographics on these instructors. We also did not collect demographics from their students. However, their students would presumably be representative of the student body at the host institution , details of which we have now included under “Participants” in the methods section.

Page 9

How did you ensure that the students were trained to code COPUS in a reliable manner? Did you calculate inter-rater reliability after this training? Please provide more details on how you tested for coder reliability.

(Please see our comment above - we used the Copus training program outlined by the authors of the instrument. We did not systematically collect IRR, but we followed the protocol until we got to agreement.)

Page 10

I was not able to see the supplemental information to be able to see the full interview protocol. Also, it would be helpful if the supplemental file information (e.g., Supplemental File S1) was noted in the text.

Thank you for noting this missing file. We will include this file in the final submission. And we have indicated it in the text.

Page 11

Who are the several researchers that thematically coded the interview responses? I think it is important to be transparent about which specific researchers did this work to build trustworthiness in the data.

We have changed the wording in the manuscript to indicate which researchers thematically coded the interview responses.

How exactly did you connect the quantitative and qualitative data using exploratory mixed methods design? It is unclear to me which COPUS and which interview data were used for these analyses. It is not until the results that I read that themes were not tied to specific reformer types. It is important that you bring up these details in the methods, not just results.

This is an excellent point. We have duplicated the statement from the results in the methods section.

Results:

Page 11

I would move this participant info to the methods section. It’s good to be transparent about the loss of participants due to COVID, but I don’t think it is a great way to start your results section.

We have adjusted the participant info in the results section.

Why did you decide to add the ranking of the instructors to the quantitative results? Are you interested in examining how instructor rankings impacted use of student-centered teaching practices? Again, I am not sure if these results are aligned to the research questions and methods as I did not clearly see research questions earlier on.

We only added this in response to a previous reviewer who requested it. We were not interested in analyzing reform by rank as we did not have enough participants in each category to really identify any meaningful patterns.

References:

Check first author spelling for this citation: Creswell, J. W., & Plano Clark, V. L. (2007). Designing and conducting mixed methods research. Thousand Oaks, CA: Sage.

We fixed it! Thank you for catching this!

---

## [Editor Report · Decision Letter 2]

3 Jul 2023

PONE-D-22-33194R2Iterating toward change: improving student-centered teaching through the STEM faculty institute (STEMFI)PLOS ONE

Dear Dr. Jensen,

Thank you for submitting your manuscript to PLOS ONE. After careful consideration, we feel that it has merit but does not fully meet PLOS ONE’s publication criteria as it currently stands. Therefore, we invite you to submit a revised version of the manuscript that addresses the points raised during the review process.

 The revised form of your article requires minimal corrections as noted in the reviewer's comments. With these arrangements, which can be completed in a short time, the article will be ready for publication.

We look forward to receiving your revised manuscript.

Kind regards,

Ayse Hilal Bati, Professor

Academic Editor

PLOS ONE

Journal Requirements:

Additional Editor Comments:

Dear Author/s,

The revised form of your article requires minimal corrections as noted in the reviewer's comments. With these arrangements, which can be completed in a short time, the article will be ready for publication. Thank you.

---

## [Editor Report · Decision Letter 3]

20 Jul 2023

Iterating toward change: improving student-centered teaching through the STEM faculty institute (STEMFI)

PONE-D-22-33194R3

Dear Dr. Jensen,

We’re pleased to inform you that your manuscript has been judged scientifically suitable for publication and will be formally accepted for publication once it meets all outstanding technical requirements.

Kind regards,

Ayse Hilal Bati, Professor

Academic Editor

PLOS ONE

Additional Editor Comments (optional):

Dear Author/s,

I am very glad that I had the opportunity to evaluate your article. You have made significant progress by improving your article with the recommendations of the referees. I think the article is suitable for publication on Plos One. Thanks
---

## [Editor Report · Acceptance letter]

9 Aug 2023

PONE-D-22-33194R3 

Iterating toward change: improving student-centered teaching through the STEM faculty institute (STEMFI) 

Dear Dr. Jensen:

I'm pleased to inform you that your manuscript has been deemed suitable for publication in PLOS ONE. Congratulations! Your manuscript is now with our production department. 

Kind regards, 

on behalf of

Dr. Ayse Hilal Bati 

Academic Editor

PLOS ONE